# Service Effectiveness of the Nature Centers for Sustainability of Environmental Education and Forest Policy Implications

**Chia-Wen Lee [1], Ching Li [2],* and Sung-Ta Liu [3],***

[1] College of Humanities and Communications, Yango University, Fuzhou 350015, China; lcwgg@mail2000.com.tw

[2] Graduate Institute of Sport, Leisure and Hospitality Management, National Taiwan Normal University, Taipei 106, Taiwan

[3] Department of Tourism, Shih-Hsin University, Taipei 116, Taiwan

* Correspondence: t94002@ntnu.edu.tw (C.L.); stliu@mail.shu.edu.tw (S.-T.L.);
  Tel.: +886-2-773-45402 (C.L.); +886-2-223-68225 (S.-T.L.)

**Abstract:** The purpose of this study was to explore the relationship between service effectiveness of the Nature Centers of Forest Bauru of Executive Yuan in Taiwan (Nature Centers) and sustainability of environmental education and forest policy implications. The (n = 1520) participants were selected through purposive and quota sampling, and the questionnaires were divided into four categories: Potential target customer groups, potential operators, the public, and website users from 1 July to 31 August 2017. Canonical correlation analyses were used to explore the relationship between two variables. The study results revealed the significant relationships between service effectiveness and forest policy implications, and two canonical factors were extracted. The first pair of canonical correlation analyses reveals that higher overall service effectiveness is more likely to catch the attentions of participants with the overall forest policy implications. The second canonical correlation analysis suggests that forest policy implications factors of protected coast forest, exotic species, green economy, forest resource management, and wood self-sufficiency ratio were positively associated with service effectiveness for natural resources and negatively associated with those of teaching quality. The implication for further research and practical applications in terms of cultural and creative research is suggested.

**Keywords:** service effectiveness; nature centers; sustainability; environmental education; forest policy implications

## 1. Introduction

Since the United Nations (UN)Commission on Sustainable Development, environmental education and forest sustainability have begun to be valued [1,2]. The governments promote the public's environmental awareness under the global initiative of environmental education [3,4]. The Nature Centers of Forest Bauru of Executive Yuan in Taiwan (Nature Centers) not only have a duty to promote sustainable forest policy, but also to comply with the concepts of service management to ensure their effective operation in order to develop sustainability of environmental education and the forest industry.

Service effectiveness refers to the efficacy of the service provided by a service provider to the customer. A provider's service comprises three dimensions, (1) service capability: The skills, abilities, and processes needed to develop close relationships with customers; (2) service approach: The process of delivering service value to customer; (3) service efficiency: Efficiency of input to output of service resource [5–8]. According to the management of outdoor recreation [9], service effectiveness should

define service capability as environmental characteristics, service approach as educational methodology, and service efficiency as operational and management process in the Nature Centers.

For the most part, service effectiveness should be evaluated by customers. The Nature Centers exist for the public good, with nonprofit-oriented objectives. When assessing their service effectiveness, the internal governance of centers must be considered for instance, how internal management measures have been applied to connect each center's objectives. Considering the objectives of Nature Centers, they should also focus on their external presentations, including external influences, competitor types, and target customer profiles, to assess service effectiveness. Therefore, this study concerned the inclusion of managers and public opinions. The concepts of environmental characteristics, educational methodology, and operations and management that comprise the service effectiveness system model are investigated and explored.

In practice, service effectiveness is an important tool for managers to positively impress the public. An interesting and an important management issue has emerged on whether managers can befittingly design the programs of environmental education to attract much more participants. The present study seeks to make three central contributions. First, it could precisely detect which factors of service effectiveness might be influenced in order to approach cost-effectiveness for existing service management literature. Second, this study provides original insight into the minds of managers and the public for evolving quality of existing programs in order to match the managing requirement to maximize influence on organization effectiveness. Finally, managers could take this study's findings to persuade organization leaders and to obtain a flow of new participants by improving their service effectiveness.

## 2. Theoretic Background

### 2.1. Service Effectiveness

Through a literature review and archive analysis of the Nature Centers' official publicity materials, a service effectiveness factor evaluation index was created containing three major and twelve corresponding items. A description of each factor follows.

#### 2.1.1. Environmental Characteristics

According to the core values of the Forest Bureau—namely "maintaining the forest ecology and conserving natural resources"—and the four major governance objectives, to establish that environmental education centers are in line with local forest characteristics and cultural resources and to actively promote sustainable forest management and natural resource conservation [10–12], the indicators for environmental analysis were as follows: (1) Natural resources: a Nature Center provides a positive interactive place between humans and nature; (2) Cultural resources: the Center provides a place to develop local culture and humanities; (3) Sustainable development: the Center provides a place for understanding the balance between human beings and natural ecology and advocates the concept of sustainability.

#### 2.1.2. Educational Methodology

With access to guided tours, commentaries, and diversified educational programs, Taiwan's citizens have started to consider and take action on environmental protection. Continuous development of civic environment education has helped achieve the goals of nature conservation and sustainable forest management [13–19]. The objectives of eight of the Nature Centers revealed that the aspects of educational methodology examined in this study were as follows: (1) Core concepts: The course participants know the relevant knowledge and issues of environmental conservation and take actions to protect the environment; (2) Curriculum programs: to combine the characteristics of the local environment and design multiple courses according to different needs; (3) Teaching quality: to make good use of the two-way communication between teachers and students, use multiple results

presentation methods, and evaluate the effectiveness of learning; (4) Academic research: To teach and to research in the field of refined nature is an indicator of educational methodology.

### 2.1.3. Operational and Management Process

The policy objectives of the Forest Bureau are to "develop recreational forest that provides high quality, promotes lifestyles of health and sustainability, and encourages in-depth tourism". They established various public facilities, including forest recreation areas and tourism industries; strategic alliances; and provided citizens with forest culture experiences. Ecotourism development and forest reserve and habitats management of the policy plans were proposed. The policy objectives of the Forest Bureau are implemented in the operations and management of the Nature Centers [20–28]. Therefore, the governance goals and objectives can be converted into operating strategies by the Center managers—through excellent management, service quality can be improved. The operations and management indicators were thus as follows: (1) Expertise: To combine internal and external experts and scholars to jointly promote sustainability of environmental education, (2) facility location: To use facilities and equipment in accordance with the concept of sustainability and carry out regular maintenance to ensure the safety of students, (3) evaluation performance: To obtain national certification of sustainability of environmental education, (4) strategic alliances: Cooperation with the government and relevant private institutions to improve the quality of sustainability of environmental education services, (5) strategic development: To promote sustainability of environmental education strategies and to share experiences with other countries so that employees can learn and be brave enough to meet challenges.

### 2.2. Environmental Education and Forest Policy Implications

The Nature Centers are affiliated with the Forest Bureau in Taiwan. Message posts released through new media should be in compliance with policies issued by the Forest Bureau and take into account environmental characteristics, educational methodology, and operational and management process of the management of outdoor recreation to ensure that the Nature Centers simultaneously engage service effectiveness under the influence of sustainability of environmental education and forest development. To achieve core administrative objectives, this study investigated the content of the 2016 policy objectives of the Forest Bureau. The objectives were: (1) to develop an ecological forest and maintain a green resource environment, (2) to develop a recreational forest industry that offers in-depth and high-quality tours with an emphasis on lifestyles of health and sustainability, (3) to develop a safe forest industry and implement the restoration and protection of national land, (4) to develop high-quality forest and forest-planting resources [29–35]. This study analyzed sixteen keywords on the basis of ecological, economic, and social dimensions, which are presented as follows:

### 2.2.1. Ecological Dimension

In terms of environmental factors, the Forest Bureau should maintain forest health and biodiversity, intensify the Satoyama Initiative, and structure the ecological network of national land to ensure sustainable ecological environment. Therefore, according to the first objective, five keywords were proposed, namely: (1) Biodiversity: To handle animal and plant conservation and habitat conservation to maintain biodiversity; (2) Soil and water conservation: To promote the conservation of water and soil resources, manage the rehabilitation of landslides, alleviate landslides and sand disasters, and ensure the safety of life and property of the surrounding objects; (3) Protected coast forest: Strengthen the construction of coastal security forest and build a green protection net on the coast; (4) Exotic species: To reduce invasive alien species against domestic ecological environment. In addition, in accordance with the second objective, the keyword (5) Friendly environment: To construct a land ecological network, promote environmentally friendly production, and form a social, productive and ecological landscape in which man and nature coexist, was proposed.

### 2.2.2. Economic Dimension

The Forest Bureau should rationally plan the "value-added" utilization of forest ecological resources to promote the sustainable development of the industry and share benefits with the public. Therefore, according to the second objective, three keywords were proposed, namely, (1) Ecotourism: To promote tourism activities that take into account natural conservation, cultural preservation and economic development; (2) Green economy: To combine local communities and industries with forest resources to drive the local green economy; (3) Co-existence and co-prosperity: To conduct diversified eco-tourism activities to promote sustainable development of local industries and economy. Furthermore, according to the third objective, four keywords were established, namely, (4) Forest resource management: Forest resource management should be defined to strengthen forest land protection and management, strengthen prohibition mechanism, and prevent the main and by-products from stealing forests; (5) Forest land management: To implement forest management in order to prevent forest land being misused; (6) Forest monitoring: To promote sustainable forest management, continuously grasp national forest resources, and establish a long-term forest resources monitoring system; (7) Wood self-sufficiency ratio: To effectively and rationally utilize plantations to provide domestic wood and bamboo timber with stable source and quantity.

### 2.2.3. Social Dimension

The Forest Bureau is responsible for improving forest land management, promoting public participation, seeking harmony and stability in the social system, and strengthening national land restoration and protection. Therefore, four keywords were proposed according to the fourth objective, namely, (1) Community interaction: To protect the shallow mountain ecosystem by interacting with local community residents and adopting friendly environmental measures; (2) Forest culture: To activate forest culture and historical sites to provide Chinese people with experience in forest culture; (3) Sustainability of environmental education: To utilize forest resources to provide high-quality places for natural learning, sustainability of environmental education and outdoor recreation; (4) The aboriginal culture: To combine with the tribes around the forest to show the concept of harmonious coexistence between aborigines and nature and traditional wisdom.

This study involved an analysis of whether the service effectiveness related with sustainability of environmental education and forest policy implication by the Nature Centers of Forest Bureau because the core values of the courses developed by the centers will be one of the crucial factors in implementing the aforementioned policies. They are also the most basic condition for the survival and development of Nature Centers. At the 2015 National Conference on Sustainable Management of Nature Center, it was proposed that when developing a curriculum, the Nature Centers should mainly utilize the characteristics of the field and the expertise of the instructors to teach forest knowledge and scientific spirit in compliance with sustainability of environmental education objectives.

## 3. Research Method

Service providers must be aware of service effectiveness management, but organizational service effectiveness should still be judged by target or potential customer groups for services to understand their performance in service effectiveness [36]. First, environmental education concerning sustainability at the Nature Centers was explored by reviewing the literature and conducting archival analysis of the centers' official publicity materials. Service effectiveness guidelines for the centers were initially constructed, and the elements corresponding to each guideline were listed. Subsequently, the expertise method was employed to examine a measure of, "the service effectiveness guidelines and appropriateness of the forest contribution" through collection of expert opinions based on aggregation of feedback from repeated questionnaires. In this process, participating experts were not informed of the identity of other experts to ensure that the insights provided were unaffected by others. To examine the appropriateness of the service performance guidelines and their factors, the experts selected

were the target segmentation of the Nature Centers. The participants were 17 scholars working in nature-related areas and loyal customers, participants in the Centers' activities and events twice per year, with mostly education-focused careers (14 teachers, two loyal customers, and one homemaker) and an average of over 15 years of work experience. More than 50% of the participants had attended activities at a Nature Center more than 10 times. Interviews were conducted from May 15 to June 15 2017. Interviews were held by phone (40 min), and the second and third rounds were conducted using electronic questionnaires. Experts' opinions achieved consistency when they had a quaternary difference of 1.00 for the appropriateness of a certain guideline or factor. All experts had no dissenting opinions for each guideline, except for 6% for "management, and performance evaluation" and 12% for "sustainable development and core concepts". This confirmed the expert validity of the "Nature Center Service Effectiveness Guidelines and Forest Policy Implications Questionnaire".

### 3.1. Participants

Participants were selected through purposive and quota sampling, and the questionnaires were divided into four categories: Potential target customer groups, potential operators, the public, and website users. The sampling methods for each category were as follows:

1. Purposive sampling was adopted for the potential target customer group questionnaire, which targeted teachers who have not hosted activities at a Nature Center in the past 2 years and university students who are currently taking teacher training courses. A total of 415 valid questionnaires were collected.
2. Purposive sampling was also adopted for the potential operator's questionnaire, which focused on operators who are operating other natural education centers, people who work in related public sector areas, relevant nonprofit organizations, university students majoring in public policy or natural resources management, and operators of certified sustainability of environmental education centers. A total of 203 valid questionnaires were collected.
3. Quota sampling was adopted for the public questionnaire, and the survey was conducted at the main railway stations of 17 counties and cities in Taiwan. Screening criteria adopted main tourist sources of the eight Nature Centers and the proportion of population in the north, middle and south regions of Taiwan. A total of 461 valid questionnaires were collected from customers of the eight Nature Centers.
4. Survey Cake was used to produce an online questionnaire for website users, the link to which was placed on the websites, Facebook pages, and blogs of the Forest Bureau, Taiwan Forest Recreation, the Forest Culture Park, and the Graduate Institute of Sport, Leisure, and Hospitality Management of National Taiwan Normal University. A total of 441 valid questionnaires were collected.

### 3.2. Procedure

From 1 July to 31 August 2017, the (n = 1520) participants were from the potential target consumers, potential managers, site filling questionnaires, website questionnaires and website managers. Participants were compensated for their time with an NTD$50 gift card and a four-color pen. The study participants were asked to provide basic demographic information—gender, age, and educational level. A pool of respondents was made up of 635 males (41.8%) and 885 females (58.2%), with896 having undergraduates' degrees (59.0%), 456 having graduates' degrees, (30.0%), 134 having senior high school degrees (8.8%), 23 having junior high degrees (1.5%), and 11 having elementary school degrees (0.7%). The average age of the participants was 36.35 year sold (SD = 12.91), 22-year-olds accounted for the most (5.3%). Participants ranged from 18 to 80 years old.

*3.3. Measures*

This study analyzed a relationship between service effectiveness of the Nature Centers, three dimensions and twelve items, and environmental education and forest policy implications, three dimension and sixteen items. (Figure 1)

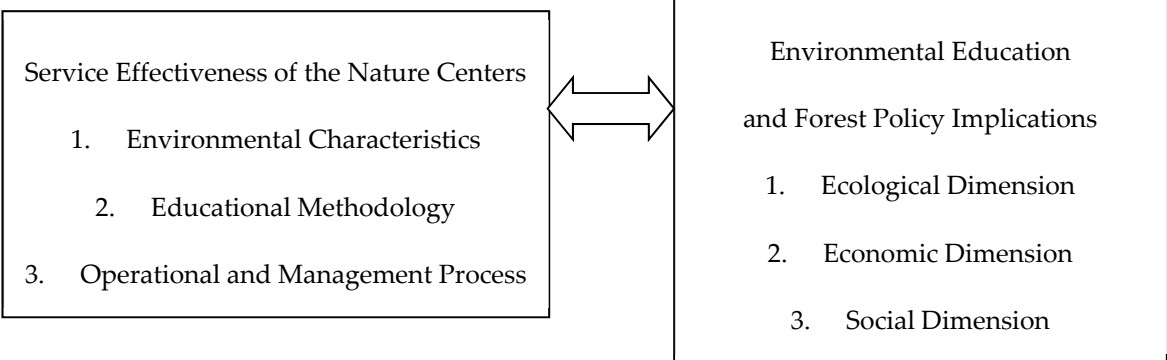

**Figure 1.** The methodological approach of this study.

3.3.1. Service Effectiveness

The service efficiency criteria of the Natural Centers of the Forest Bureau can be discussed from three dimensions. The results show that the highest average degree of education characteristics of the sample is 5.96. The lowest average for operation and management was 5.64. The highest average degree of sample identification is 5.99 for natural resources. The lowest average recognition level was teaching quality, of 5.66. The average scores of other items are: sustainable development (5.94), core concept (5.92), curriculum programs (5.83), expertise (5.82), facility location, evaluation performance (5.78), strategic alliance (5.77), academic research (5.73), and strategic development and culture resources, with an average of 5.71. Participants rated the score on a 7-point Likert-type scale ranging from strongly disagree (1) to strongly agree (7). Overall, these items exhibited satisfactory internal reliability, with Cronbach's αwas from 0.96–0.97, and thus the scale showed a high degree of reliability.

3.3.2. Forest Policy Implications

The influence of the Nature Centers on forest policy was assessed by asking participants how they felt about their contribution according to sixteen indicators at three dimensions. The results showed that the highest average score of sustainability of environmental education was 5.81. The lowest average wood self-sufficiency ratio was 5.24. The other scores were: ecotourism (5.66), biodiversity (5.62), forest culture, coexistence and co-prosperity (5.55), friendly environment (5.52), forest monitoring (5.50), community interaction with 5.47, the aboriginal culture, and green economy, soil and water conservation with 5.45, followed by forest land management (5.42), and 5.37 for forest resource management, 5.36 for protected coast forest, and finally exotic species (5.29). The statement was scored from 1 point for "very insignificant" and 7 for "very significant". Cronbach's α value was ranged from 0.96–0.98.

*3.4. Data Analysis*

Data were assessed for frequencies and percentages on each item. A set of descriptive, Pearson's correlations, and canonical correlation analyses was employed to evaluate the study objective. The study's use of canonical correlation analysis was appropriate, since it measures the overall relationship between two sets of variables. This study attempted to unveil an association between these variables of service effectiveness and forest policy implications.

## 4. Results

The participants had to indicate the evaluation of service effectiveness and forest policy implications. The participants' evaluation of service effectiveness was assessed on twelve items: Borrowed natural resources, cultural resources, sustainable development, core concepts, curriculum programs, teaching quality, academic research, expertise, facility location, evaluation performance, strategic alliance, and strategy development. The scale of forest policy implications was analyzed using factor analysis run over the sixteen items: Sustainability of environmental education, wood self-sufficiency ratio, ecotourism, biodiversity, forest culture, coexistence and co-prosperity, friendly environment, forest monitoring, community interaction, the aboriginal culture, green economy, soil and water conservation, forest land management, forest resource management, protected coast forest, and exotic species. The factor loadings give us an idea about how much the variable has contributed to the factor—the larger the factor loading, the more the variable has contributed to that factor. The factors with eigenvalues equal to or greater than 1 are included in the factor model, and their factor loadings are calculated [37]. Each factor had an eigenvalue above 1.0, the factors with eigenvalues equal to or greater than 1 are included in the factor model, and their factor loadings are calculated. Cronbach's $\alpha$ for these items' scale was ranged from 0.96 to 0.98. In terms of the reliability of the test, both the two measures of service effectiveness guidelines and appropriateness of the forest contribution revealed that can provide high dependable and consistent information. In terms of the content validity of the test, several rounds of expertise evaluation were conducted to confirm that the measure had good content validity, the measures had to reflect all of these aspects.

To understand the correlation among the different variables, this study evaluated if there was a relationship between service effectiveness and forest policy implications. The results revealed that two canonical correlations were significant. Using the cutoff correlations of 0.3 to select variables for each variable set, the variables in the set of service effectiveness that was correlated with the first canonical variate were natural resources, cultural resources, sustainable development, core concepts, curriculum programs, teaching quality, academic research, expertise, facility location, evaluation performance, strategic alliance, and strategy development. The variable set of forest policy implications for sustainability of environmental education, wood self-sufficiency ratio, ecotourism, biodiversity, forest culture, coexistence and co-prosperity, friendly environment, forest monitoring, community interaction, the aboriginal culture, green economy, soil and water conservation, forest land management, forest resource management, protected coast forest, and exotic species was correlated with the first canonical variate. The first pair of canonical variates revealed that service effectiveness for natural resources (0.80), humanistic characteristics (0.79), sustainable development (0.82), core idea (0.80), curriculum plan (0.82), teaching quality (0.82), academic research (0.88), professional talents (0.86), facility location (0.81), evaluation performance (0.84), strategic alliance (0.90), and strategy development (0.90) was positively associated with forest policy implications for biodiversity (0.89), soil and water conservation (0.77), protected coast forest (0.76), exotic species (0.72), friendly environment (0.84), forest monitoring (0.89), ecotourism (0.89), green economy (0.86), coexistence and co-prosperity (0.87), forest resource management (0.75), forest land management (0.72), wood self-sufficiency ratio (0.74), community interaction (0.87), forest culture (0.86), sustainability of environmental education (0.86), and the aboriginal culture (0.84). There were specially higher positive relationship between the factors of academic research, professional talents, strategic alliance, and strategy development, and the factors of biodiversity, ecotourism, green economy, coexistence and co-prosperity, and community interaction, forest culture, and sustainability of environmental education.

The second canonical variate suggested that service effectiveness for natural resources (−0.37) was positively associated with forest policy implications for environmental (−0.33) and negatively associated with forest policy implications for protected coast forest (0.23), exotic species (0.29), green economy (0.31), forest resource management (0.23), wood self-sufficiency ratio (0.33) as well as service effectiveness for teaching quality (0.26), contrary to the natural resources. The factors of natural resources and teaching quality had a bidirectional connection with the factors of protected coast forest,

exotic species, green economy, forest resource management, and wood self-sufficiency ratio. Table 1 summarizes the findings.

**Table 1.** Canonical Correlation Analysis between Service Effectiveness and Forest Policy Implications.

| Variable X | Canonical Variables | | Variable Y | Canonical Variables | |
|---|---|---|---|---|---|
| | X1 | X2 | | Y1 | Y2 |
| Natural resources | 0.80 | −0.37 | Biodiversity | 0.89 | −0.14 |
| Cultural resources | 0.79 | 0.20 | Soil and water conservation | 0.77 | 0.13 |
| Sustainable development | 0.82 | −0.20 | Protected coast forest, | 0.76 | 0.23 |
| Core concepts | 0.80 | −0.16 | Exotic species | 0.72 | 0.29 |
| Curriculum programs | 0.82 | −0.13 | Friendly environment | 0.84 | 0.17 |
| Teaching quality: | 0.82 | 0.26 | Forest Monitoring | 0.79 | 0.03 |
| Academic research | 0.88 | 0.11 | Ecotourism | 0.89 | −0.12 |
| Expertise | 0.86 | −0.05 | Green economy | 0.86 | 0.31 |
| Facility location | 0.81 | 0.02 | Coexistence and co-prosperity | 0.87 | 0.04 |
| Evaluation performance | 0.84 | −0.17 | Forest resource management | 0.75 | 0.23 |
| Strategic alliances | 0.90 | 0.10 | Forest land management | 0.72 | 0.15 |
| Strategic development | 0.90 | 0.11 | Wood self-efficiency | 0.74 | 0.33 |
| | | | Community interaction | 0.87 | 0.15 |
| | | | Forest culture | 0.86 | −0.09 |
| | | | Environmental education | 0.86 | −0.33 |
| | | | The aboriginal culture | 0.84 | 0.23 |
| The percentage of variance | 61.91 | 12.27 | The percentage of Variance | 72.68 | 10.93 |
| Overlap | 25.14 | 1.02 | Overlap | 29.52 | 30.43 |
| $p^2$ | | 0.56 | 0.15 | | |
| Canonical correlation | | 0.75 | 0.39 | | |
| $p$ | | 0 | 0 | | |

## 5. Discussion and Conclusion

This study attempts to provide insights into the relatively unfamiliar nature of forest policy implications and service effectiveness, specifically in terms of service effectiveness and how the factors of service effectiveness relate to those of forest policy implications. The results show that two pairs of canonical correlation variates are significant in the relationship between service effectiveness and forest policy implications.

### 5.1. Operator Viewpoints and Management Prospects

In terms of the relationship between the set of service effectiveness and the set of forest policy implications variables, the first pair of canonical correlation analyses reveal that higher overall service effectiveness is more likely to catch the attentions of participants with the overall forest policy implications. The findings imply that forest practitioners could strategically strengthen the Nature Centers' service effectiveness to stimulate exogenously the effectiveness of service management in order to achieve maximum forest policy implications. Five major stages were found in terms of Nature Centers' professional contributions to forest and environmental education: (1) Focus on

the cultivation of environmental education talents, (2) creating a quality natural education field that emphasizes learning from nature, (3) strengthening experience exchange and accumulating professional capacity, (4) promoting multiple learning channels and enhancing the influence of environmental education, (5) integrating resources and experiences and formulating operating connotations to improve environmental learning strategies. Value and recognition of the degree of service effectiveness differed between operators and the public. However, in response to the development of Nature Centers, operators have gradually shifted the focus of service effectiveness from cultivation of professional talents to curriculum planning and evaluation of academic research and development, and then to strategic alliances (local community and nongovernmental organizations) and developments (international communication and paradigm shifts). This outcome is consistent with the typical first set of linearity in correlation analysis—that is, service effectiveness as focused on by Nature Center operators helped the public understand the contribution of the Nature Centers to forest policy and environmental education. Nonetheless, differences in value and recognition of service effectiveness exist between operators and the public, and, therefore, operators should focus on service effectiveness and raising awareness of policies to eliminate differences in perception between operators and the public.

*5.2. Diversified Sustainability Environmental Education Implicates Forest Policy*

Interestingly, the second canonical correlation analysis suggests that forest policy implication factors of protected coast forest, exotic species, green economy, forest resource management, and wood self-sufficiency ratio were positively associated with service effectiveness for natural resources and negatively associated with those of teaching quality. The service efficiency of natural resources is an important base of sustainability of environmental education, and it also shows the importance of teaching quality to forest policy promotion. Teaching quality depends on the design and implementation of human resources, and activities and diversification of the curriculum. The main axis of the management of Natural Centers is based on the service of the course teaching program. Specifically, apart from relying on professional teaching staff and design and implementation of curriculum activities, quality teaching requires a diverse curriculum. Successfully managing a nature education center requires teaching programs, and forest-related teaching programs may therefore help centers in promoting forest policies. Moreover, when recruiting or cultivating talents, the compatibility of human resources and policies should be considered to ensure that developed curriculums are properly integrated with the governance directions of the Forest Bureau. The implication for further research and practical applications in terms of cultural and creative research is suggested.

## 6. Limitation and Future Research

Environmental education and forest policy implications should be evaluated long-term, but this study applied the cross-sectional investigation during 2017. Future research should concern longitudinal research design.

On the other hand, this study conducted quota sampling by classification and screening of expertise meetings. Although quota sampling had the advantage of cost-effectiveness and being easy to account for population proportion in this study, the sample was not chosen by random selection and had a potential for bias. Thus, the consideration of selection weights of sample should be warranted in future in order to improve sample accuracy.

**Author Contributions:** Supervision, C.L.; Writing—original draft, C.-W.L.; Writing—review & editing, S.-T.L.

**Acknowledgments:** The authors would like to thanks the Forest Bauru of Executive Yuan, Taiwan under contract No. tfbc-1060303. This manuscript was translated into English by Wallace.

**Conflicts of Interest:** The authors declare no conflict of interest.

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
