# Peer review of "Service Effectiveness of the Nature Centers for Sustainability of Environmental Education and Forest Policy Implications"

_sustainability, doi:10.3390/su11092457_

Round 1
Reviewer 1 Report
The paper is interesting and deals with e relevant topic. However, I have some general remarks:
1.Introduction-Background could be improved with the introduction of more international references.
For instance L54 “From previous studies, service effectiveness should be evaluated by customers”-What studies?
2. Theoretic background- It is limited and it doesn’t present international studies related to this work in detail, providing a more a more complete theoretical background. They seem to focus on the description of the : factors and respective references of a service effectiveness factor evaluation index.
L36- “A nature centers” should be “A nature center”
“3. Research Mothed”-should be “3. Research Method”.
3. Research Method- I recommend providing a graphical representation of the methodological approach and it’s implementation. The sampling process and the questionaires implementation seem to be OK.
4.results-The presentation of the results is OK.
Discussion-It needs to be more detailed.
There are several English errors and of writing. Therefore the text should be revised further.
Author Response
Dear Reviewer;
Your comment made my paper complete.
Thank you for your assistance.
Revision plan is following:
Introduction-Background could be improved with the introduction of more international references.
Added
For instance L54 “From previous studies, service effectiveness should be evaluated by customers”-What studies?
Revised
Theoretic background- It is limited and it doesn’t present international studies related to this work in detail, providing a more a more complete theoretical background. They seem to focus on the description of the : factors and respective references of a service effectiveness factor evaluation index.
Added and revised
L36- “A nature centers” should be “A nature center”
Revised
Research Mothed”-should be “3. Research Method”.
Revised
Research Method- I recommend providing a graphical representation of the methodological approach and it’s implementation. The sampling process and the questionaires implementation seem to be OK.
Draw a graphic of method of approach
Results-The presentation of the results is OK.
Discussion-It needs to be more detailed.
There are several English errors and of writing. Therefore the text should be revised further.
Added and revised.
Best regards,
Prof Lee

Reviewer 2 Report
Congratulation but certain changes noted in the file should be done

Author Response
Dear Reviewer;
Your comment made my paper completed.
Thank you for your kind assistance.
Attached a revised paper.
Best regards,
Prof. Lee

Round 2
Reviewer 1 Report
The authors careful replied to my remarks. The paper is ready for publication now.